# Expressiveness of Graph Neural Networks in Planning Domains

**Primary Keywords:** *(2) Learning;*

## Abstract

Graph Neural Networks (GNNs) have recently become the standard method of choice for learning with structured data, demonstrating particular promise in classical planning. Their inherent invariance under symmetries of the input graphs endows them with superior generalization capabilities, compared to their symmetry-oblivious counterparts. However, this comes at the cost of limited expressive power. Particularly, it is known that GNNs cannot distinguish between graphs that satisfy identical sentences of $C_2$ logic.

To leverage GNNs for learning policies in PDDL domains, one needs to encode the contextual representation of the planning states as graphs. The expressiveness of this encoding, coupled with a specific GNN architecture, then hinges on the absence of indistinguishable states necessitating distinct actions. This paper provides a comprehensive theoretical and statistical exploration of such situations in PDDL domains across diverse natural encoding schemes and GNN models.

## 1 Introduction

Deep learning Neural Networks (NNs) occupy a prominent position in contemporary artificial intelligence (AI) research, exerting a substantial influence over all AI domains. Their recent generalization to variable-sized structured data, in the form of GNNs, then enabled their integration into classical planning, too (Toyer et al. 2018, 2020; Ståhlberg, Bonet, and Geffner 2022a,b, 2023). The focus of this paper is on their use for learning policies in PDDL domains. More precisely, given a PDDL domain, we consider a policy-learning algorithm that utilizes an NN model to predict the next action to take in any state that belongs to a given domain instance.

Two aspects decide whether the policy-learning algorithm successfully solves a PDDL domain. The first aspect is the learning capacity of the underlying NN model, reflected mostly by its size, which must adequately reflect the combinatorial complexity of solving the given domain instances. The second aspect is the *expressiveness* of the NN model that must be sufficient for the given domain. The central aim of this paper is to focus on the latter. This is important since the structured NN architectures, such as GNNs, lack the universal approximation capabilities of their classic feed-forward counterparts. GNNs represent functions whose output is invariant under the (permutation) symmetries of its inputs, which significantly improves their learning generalization. However, this comes at the cost of limited expressive power. Particularly, GNNs cannot distinguish between graphs beyond the constraints imposed by the 1-dimensional Weisfeiler-Lehman (WL) test (Morris et al. 2019; Xu et al. 2019; Grohe 2021). We believe this aspect should be investigated in separation to discern whether a model is insufficient because of its limited learning capacity, or insufficient expressive power.

There are two possible approaches to developing an NN policy-learning algorithm, either (i) design a custom, specifically tailored NN model, or (ii) use one of the existing GNN models applied on top of a suitable encoding, transforming domain instances into a graph-like data structure. An example of the first approach is ASNets (Toyer et al. 2018, 2020), specifically crafted to reflect the structure of a planning domain and its instances, namely their actions and facts. This class of custom approaches comes with considerable variability but that also makes it difficult to systematically study its expressiveness. This is why we focus on the second approach, building on the existing, widespread GNN architectures (Wu et al. 2020) and their associated expressiveness characterizations (Grohe 2021). The schema of this approach, with example works including (Ståhlberg, Bonet, and Geffner 2022a,b, 2023), is depicted in Fig. 1. Given a PDDL domain, its instance, and its particular state, respectively, we encode the corresponding contextual state representation into a suitable data structure to be processed by a GNN to output an action prediction.

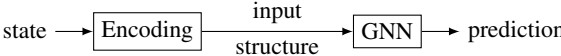

Figure 1: Procedural flow of the selected approach

The encoding, together with the GNN architecture, then determines the overall expressiveness of the resulting policy-learning algorithm. This integration establishes a function $f$ which, for a given state $s$ and GNN parameters $\theta$, yields a prediction $f(s, \theta)$. The expressiveness of this function then lies in its ability to distinguish between different states. Particularly, we call two states $s, s'$ *indistinguishable* w.r.t. $f$ if $f(s, \theta) = f(s', \theta)$ for *any* tuple of parameters $\theta$. Note that this is a desired property of all the

structured, symmetry-aware (G)NN architectures. However, we call an indistinguishable pair *bad* if $s, s'$ should lead to different predictions. The main contribution of this paper is a theoretical and statistical analysis probing the prevalence of such bad state pairs in PDDL domains.

## 2 Background

### First-Order Logic

We recall several concepts from mathematical logic that we will need in the sequel. A *first-order relational language $\mathcal{L}$* consists of predicate symbols together with their respective arities and variables.[1]

An *atomic formula* is an expression of the form $R(x_1, \ldots, x_n)$ where $R$ is an $n$-ary predicate symbol and each $x_i$ is a variable. The formulas of first-order logic (FOL) are built from atomic formulas using Boolean connectives and quantifiers $\forall, \exists$. A FOL-formula without free variables (i.e., all its variables belong to the scope of a quantifier) is called a *sentence*.

The FOL formulas are interpreted in $\mathcal{L}$-structures. An $\mathcal{L}$-*structure* is a tuple $\mathbf{S} = \langle S, \langle R^{\mathbf{S}} \mid R \in \mathcal{L} \rangle \rangle$ where $S$ is a set of objects, and $R^{\mathbf{S}} \subseteq S^n$ is the interpretation of the $n$-ary predicate symbol $R \in \mathcal{L}$. The fact that a FOL-sentence $\varphi$ holds in $\mathbf{S}$ is denoted $\mathbf{S} \models \varphi$.

A *(ground) atom* over a set of objects $S$ is an expression of the form $R(b_1, \ldots, b_n)$ where $R$ is an $n$-ary predicate symbol and each $b_i \in S$. The ground atom $R(b_1, \ldots, b_n)$ holds in $\mathbf{S}$ iff $\langle b_1, \ldots, b_n \rangle \in R^{\mathbf{S}}$. Note that there is a one-to-one correspondence between $\mathcal{L}$-structures on $S$ and sets of ground atoms over $S$. Namely, an $\mathcal{L}$-structure $\mathbf{S}$ can be represented as the set of ground atoms $\psi^{\mathbf{S}}$ that hold in $\mathbf{S}$, and conversely a set of ground atoms $\psi$ represents an $\mathcal{L}$-structure $\mathbf{S}_\psi$ whose relation $R$ is defined by $\vec{c} \in R^{\mathbf{S}_\psi}$ iff $R(\vec{c}) \in \psi$.

We say that two $\mathcal{L}$-structures $\mathbf{S}, \mathbf{S}'$ are *isomorphic* if there is a bijection $\sigma \colon S \to S'$ such that $\vec{c} \in R^{\mathbf{S}}$ iff $\sigma(\vec{c}) \in R^{\mathbf{S}'}$ for each predicate $R \in \mathcal{L}$. In the expression $\sigma(\vec{c})$, the bijection $\sigma$ is applied component-wise to each object in $\vec{c}$.

We call an $\mathcal{L}$-structure *binary* if all its relations are at most binary. A *graph* is a binary $\mathcal{L}$-structure $\mathbf{G} = \langle G, E^{\mathbf{G}}, P_1^{\mathbf{G}}, \ldots, P_m^{\mathbf{G}} \rangle$ for the language $\mathcal{L}$ consisting of a binary predicate symbol $E$ representing edges, and several unary predicate symbols $P_1, \ldots, P_m$ representing node properties. As graphs are not directed, $E^{\mathbf{G}}$ must be a symmetric relation. Given $\mathbf{G}$ and a node $u \in G$, the set of its neighbours is denoted $N^{\mathbf{G}}(u) = \{v \in G \mid \langle u, v \rangle \in E^{\mathbf{G}}\}$.

The *logic C* is an extension of FOL, adding counting existential quantifiers $\exists^{\geq n}$ for each $n \geq 1$. A sentence $\exists^{\geq n} x \varphi(x)$ holds in a structure if there are at least $n$ pairwise different objects $b$ such that $\varphi(b)$ holds. The counting quantifiers are definable in FOL but they are not if we restrict the number of variables. The *logic $C_k$* is the fragment of C having only $k$ variables. To express more complex properties of a structure like a graph in $C_k$, one often needs to "reuse"

---

[1] For simplicity, we assume that $\mathcal{L}$ do not contains constants. They can be easily simulated by unary predicates. More precisely, for each constant $c$, one introduces a unary predicate symbol $P_c$ such that $P_c(x)$ holds iff $x = c$.

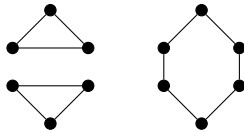

Figure 2: Two example $C_2$-equivalent graphs

variables in nested formulas. For instance, in $C_2$ one can express that there is a node in a graph with property $P_1$, such that all of its neighbours have at least three neighbours without property $P_2$, as follows:

$$\exists x P_1(x) \wedge \forall y (E(x, y) \to \exists^{\geq 3} x (E(y, x) \wedge \neg P_2(x)))$$

The last quantifier $\exists^{\geq 3}$ reuses the variable $x$. Inside the scope of $\exists^{\geq 3}$, we cannot refer to $x$ introduced by the first existential quantifier anymore. Due to this limitation, $C_2$ can express that there is a path in a graph of any length, but not a cycle of length greater than 2, since once we reuse a variable we cannot refer back to check whether we closed the cycle.

We say that two $\mathcal{L}$-structures $\mathbf{S}, \mathbf{S}'$ are $C_k$-*equivalent* if they satisfy the same $C_k$-sentences, i.e., for each $C_k$-sentence $\varphi$ we have $\mathbf{S} \models \varphi$ iff $\mathbf{S}' \models \varphi$. Two example non-isomorphic $C_2$-equivalent graphs are depicted in Fig. 2.

### Classical Planning

We assume that the reader is familiar with classical planning and Planning Domain Definition Language (PDDL). We consider the normalized, non-numeric, non-temporal PDDL tasks without conditional effects, axioms, and negative preconditions, and with all formulas being conjunctions of atoms. The types are modeled as unary predicates. Hence, for each type (i.e., a set of objects), a corresponding unary predicate is interpreted by that set of objects; for details see (Helmert 2009).

A *PDDL Domain* is a pair $\mathcal{D} = \langle \mathcal{L}, \mathcal{A} \rangle$ where $\mathcal{L}$ is a first-order relational language and $\mathcal{A}$ is a collection of *action schemas*. A *state* in $\mathcal{D}$ is an $\mathcal{L}$-structure $\mathbf{S}$. The action schemas define a state transition system using grounding in the usual way (Corrêa et al. 2020). Since we only deal with states, we skip the formal definition of the transition system.

Given a state $\mathbf{S}$, a *goal* is a set of ground atoms built from objects in $S$. A *goal state* is a state where all atoms from $\psi_G$ hold. A *planning instance* for a domain $\mathcal{D}$ is a pair $\langle \mathbf{S}, \psi_G \rangle$ where $\mathbf{S}$ is a state and $\psi_G$ is a goal. Following (Ståhlberg, Bonet, and Geffner 2022a), we integrate the goal $\psi_G$ into the state $\mathbf{S}$ by expanding its language $\mathcal{L}$. We define an expansion $\mathcal{L}_G$ of $\mathcal{L}$ such that for each $n$-ary predicate $R$ in $\mathcal{L}$, there is a new $n$-ary predicate $R_G$ in $\mathcal{L}_G$. Given a state $\mathbf{S}$ and a goal $\psi_G$, an *enriched state* $\mathbf{S}_G$ is an $\mathcal{L}_G$-structure where $R^{\mathbf{S}_G} = R^{\mathbf{S}}$ for each predicate $R \in \mathcal{L}$, and $R_G^{\mathbf{S}_G} = \{\vec{c} \mid R(\vec{c}) \in \psi_G\}$ for each predicate $R_G \in \mathcal{L}_G \setminus \mathcal{L}$. From now on, when we refer to a state, we mean a state enriched by the goal information.

A *plan* for a state $\mathbf{S}_G$ is a sequence of ground actions transforming $\mathbf{S}$ into a goal state. A plan is called *optimal* if it is shortest among all plans.

The optimal value function $V^*$ assigns to a state the length of its optimal plan, and $\infty$ if there is none. The greedy policy

for $V^*$, selecting a successor state with the minimum value of $V^*$, is the optimal policy for the domain $\mathcal{D}$.

To make the GNN predictions concrete in our experiments, we assume that the GNN predicts the optimal value function $V^*$ that defines the optimal greedy policy, in the same way as (Ståhlberg, Bonet, and Geffner 2022a).

The following theorem ensures that we can represent $V^*$ by a model that is invariant under symmetries of its inputs.

**Theorem 1.** *Let* $\mathcal{D} = \langle \mathcal{L}, \mathcal{A} \rangle$ *be a PDDL domain and* $\mathbf{S}, \mathbf{S}'$ *two isomorphic* $\mathcal{L}_G$-*structures. Then* $V^*(\mathbf{S}) = V^*(\mathbf{S}')$.

The theorem can be proven by induction on the length of the plan. From Theorem 1, it also follows that states with different values of $V^*$ can be distinguished by a FOL sentence. Unfortunately, this need not be true for $C_2$-sentences, as we will see in the next sections.

## Graph Neural Networks

GNNs have recently become the standard architecture for structured data, applied across various tasks in the form of node, edge, and graph classification/regression, with the latter corresponding to the prediction of $V^*$ that we consider in this paper. GNN models generally take as input a *binary* graph-like structure whose nodes, and possibly edges, are labeled by real-valued *"feature vectors"*. These are then consecutively processed through a series of parameterized differentiable transformations into the output prediction value.

The specific GNNs may differ in many aspects. The relevant one for this paper is their input format, which can be an actual graph but also an arbitrary binary structure. If it is a graph, we only have a single undirected edge relation, and all the nodes must be of the same type, reflected by shared feature vector dimensions. If it is a binary structure, we can have several directed edge relations, corresponding to a *multi-graph*. Moreover, if the nodes are of different types, reflected by differently sized feature vectors, we call the input a heterogeneous (multi) graph, with a special case being the *bipartite* graph where we only have two types of nodes. Independently from the above classification, a GNN architecture either supports edge features or not.

GNN models are generally based on the concept of "message-passing". From the theoretical point of view, the expressiveness of message-passing architectures has been extensively studied for graphs without edge features, as defined in the FOL Section 2. We recall the most important results here; for an overview see (Grohe 2021).

Given a graph $\mathbf{G} = \langle G, E, P_1, \ldots, P_n \rangle$, the feature vector $\boldsymbol{u} \in \{0, 1\}^n$ of a node $u \in G$ is constructed from the unary predicates $P_i$ as follows: $\boldsymbol{u}_i = 1$ iff $P_i(u)$ holds in $\mathbf{G}$. We refer to the above construction as a *multi-hot encoding*.

A GNN $g$ based on the message passing scheme consists of several layers $\ell_1, \ldots, \ell_k$. Each layer $\ell_i$ for $i < k$ takes a graph $\mathbf{G}$ whose nodes are labeled by a real-valued vector and updates its labeling. For a node $u \in G$, we denote the feature vector entering the first layer by $\boldsymbol{u}^{(0)}$. Each layer $\ell_i$ for $i < k$ then produces new vectors $\boldsymbol{u}^{(i)}$ based on the formula:

$$\boldsymbol{u}^{(i+1)} = \mathsf{comb}(\boldsymbol{u}^{(i)}, \mathsf{agg}(\{\!\{\boldsymbol{v}^{(i)} \mid v \in N^{\mathbf{G}}(u)\}\!\})) \quad (1)$$

where $\{\!\{\ldots\}\!\}$ denotes a multiset, $\mathsf{agg}$ is a function aggregating the multiset of the feature vectors of all neighbours of a node $u$, and $\mathsf{comb}$ is a function combining the feature vector of $u$ with the result of $\mathsf{agg}$.

The final "read-out" layer $\ell_k$ then takes the multiset of all the vectors from the $\ell_{k-1}$ layer and, similarly to $\mathsf{agg}$, aggregates them into a single output value:

$$\mathsf{ro}(\{\!\{\boldsymbol{u}^{(k-1)} \mid u \in G\}\!\}) \quad (2)$$

We denote the tuple of all the parameters contained within the respective comb, agg, and ro parameterized differentiable functions from all the layers of the GNN $g$ as $\theta$, and the output of such $g$ as $g(\mathbf{G}, \theta)$.

Note that the computation done by the message-passing is conceptually the same as in the Color Refinement Algorithm (CR). CR computes a node-coloring for a colored graph by applying Equation (1), with $\mathsf{agg}$ being the identity function and comb being concatenation. CR can be used as an isomorphism test for two graphs that is as strong as the 1-dimensional Weisfeiler-Lehman test (1-WL); see (Grohe 2021). It is known that 1-WL cannot distinguish all non-isomorphic graphs. More precisely, it can distinguish exactly those that are not $C_2$-equivalent (Cai, Fürer, and Immerman 1992). Thus, it is not surprising that message-passing GNNs have the same limitations. The following two theorems were independently proved in (Morris et al. 2019; Xu et al. 2019).

**Theorem 2.** *Let* $g$ *be a GNN and* $\mathbf{G}, \mathbf{G}'$ *two graphs. If* $\mathbf{G}, \mathbf{G}'$ *are* $C_2$-*equivalent, then* $g(\mathbf{G}, \theta) = g(\mathbf{G}', \theta)$ *for any tuple of parameters* $\theta$.

It follows from the above theorem that we cannot distinguish two $C_2$-equivalent graphs by a message-passing GNN, no matter its number of layers or dimensionality of the parameters used in their comb and agg functions.

The reverse direction holds as well, even though the result is not uniform. Particularly, for each $n$ there is a GNN that is able to distinguish all pairwise $C_2$-non-equivalent graphs up to size $n$.

**Theorem 3.** *Let* $n \geq 1$. *There is a GNN* $g$ *and parameters* $\theta$ *such that for all graphs* $\mathbf{G}, \mathbf{G}'$ *of order at most* $n$, *if* $\mathbf{G}, \mathbf{G}'$ *are not* $C_2$-*equivalent, then* $g(\mathbf{G}, \theta) \neq g(\mathbf{G}', \theta)$.

Whether a particular GNN actually distinguishes two graphs depends on the number of layers and the functions comb, agg and ro. Crucially, it depends on how well agg can separate two different multisets of (feature) vectors. For instance, it is known that agg based on the summation sum are more expressive than those based on mean, with max being the least expressive variant (Xu et al. 2019).

To further increase the GNN expressiveness, there are various *higher-order* GNN variants, mostly mimicking the $k$-dimensional WL test; see (Grohe 2021). Here, instead of single nodes, we compute the 1-WL over a graph built from all $k$-*tuples* of nodes. Such variants can then distinguish graphs up to $C_k$-equivalence. However, the number of $k$-tuples grows fast with $k$, hence variants based on $k$-*sets* (sets of size $k$) were introduced instead (Morris et al. 2019). Although $k$-sets are more efficient, they lose the theoretical connection with the WL test.

## 3 Encodings

To apply a GNN to learn a policy, one needs to design an encoding of the enriched state into a data structure compatible with the GNN's input format. In this section, we systematically explore the space of such possible encodings to be later tested in our experiments. Each encoding converts an $\mathcal{L}$-structure $\mathbf{S}$ either to a labeled graph, multi-graph, or bipartite (multi) graph (Sec. 2). We further decompose each encoding into two steps. The first step converts the $\mathcal{L}$-structure $\mathbf{S}$ to a binary structure $\mathbf{S}_2$. The second step transforms the binary structure into the GNN input.

To define a binary structure $\mathbf{S}_2$ from an $\mathcal{L}$-structure $\mathbf{S}$, there are several natural choices for creating its objects:

- the objects in $\mathbf{S}_2$ are the *objects* in $\mathbf{S}$,

- the objects in $\mathbf{S}_2$ are the *atoms* that hold in $\mathbf{S}$,

- the combination of the above cases, i.e., we create objects in $\mathbf{S}_2$ from *both* the objects and the valid atoms in $\mathbf{S}$,

- the objects in $\mathbf{S}_2$ are (ordered) *pairs* of objects from $\mathbf{S}$.

In the following subsections, we discuss each variant in detail. As a running example illustrating each encoding, we use the $\mathcal{L}_G$-structure from the following example.

**Example 4.** Consider a language $\mathcal{L}$ consisting of a nullary predicate symbol $N$, two unary predicates $R, L$, a binary $A$, and ternary $T$. Recall that the language $\mathcal{L}_G$ is the extension of $\mathcal{L}$ by the goal copies of all the predicates. The $\mathcal{L}_G$-structure $\mathbf{S}$ is defined over the set of objects $S = \{r, l_1, l_2\}$ so that it satisfies exactly the following set of ground atoms:

$$\{N, R(r), L(l_1), L(l_2), A(r, l_1), A_G(r, l_2), T(r, l_1, l_2)\}$$

Before we describe the particular encodings, we introduce our notation regarding tuples of objects. For a tuple of objects $\vec{c}$, $c_i$ denotes its $i$-th element. Given a set of objects $B$ and a tuple of objects $\vec{c}$, we use the expression $B \subseteq \vec{c}$ to state that each object from $B$ occurs in $\vec{c}$. For two tuples $\vec{b}, \vec{c}$, the expression $\vec{b} \cap \vec{c}$ denotes the set of their shared objects. Analogously, $\vec{b} \cup \vec{c}$ is the set of all objects occurring in at least one of the tuples. Finally, $\vec{b} \triangle \vec{c} = (\vec{b} \cup \vec{c}) \setminus (\vec{b} \cap \vec{c})$ is the symmetric difference of the elements occuring in $\vec{b}$ and $\vec{c}$.

### Object Binary Structure

The first encoding keeps the sets of objects in $\mathbf{S}$ and $\mathbf{S}_2$ the same. To reduce the arity of predicates greater than 2 to binary, we simply relate two objects if they jointly occur in an atom that holds in $\mathbf{S}$. Given a language $\mathcal{L}$, we define its binarization $\mathcal{L}_O$ as the language consisting of the same predicate symbols as $\mathcal{L}$, but with arity of all the predicates beyond binary being equal to 2.

**Definition 5.** Let $\mathbf{S} = \langle S, \langle R^{\mathbf{S}} \mid R \in \mathcal{L} \rangle \rangle$ be an $\mathcal{L}$-structure. We define its *object binary structure* as an $\mathcal{L}_O$-structure $\mathbf{S}_2 = \langle S, \langle R^{\mathbf{S}_2} \mid R \in \mathcal{L}_O \rangle \rangle$ where $R^{\mathbf{S}_2} = R^{\mathbf{S}}$ for a nullary or unary predicate symbol $R$. For a predicate $R$ of arity at least 2, we define

$$\langle c_1, c_2 \rangle \in R^{\mathbf{S}_2} \text{ iff } \{c_1, c_2\} \subseteq \vec{c} \text{ for some } \vec{c} \in R^{\mathbf{S}}.$$

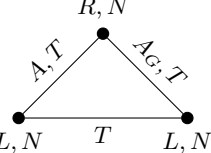

Figure 3: Object binary structure $\mathbf{S}_2$ built from the $\mathcal{L}_G$-structure introduced in Example 4.

Note that all the binary relations in $\mathbf{S}_2$ are symmetric by the definition because $c_1, c_2$ might occur in any position in $\vec{c}$. As a variant, it makes sense to keep the binary relations in $\mathbf{S}_2$ the same as in $\mathbf{S}$. However, for relations $R^{\mathbf{S}}$ of arity greater than 2, there is no canonical way to define directions on $R^{\mathbf{S}_2}$. Hence, we keep all the binary relations symmetric in $\mathbf{S}_2$ for simplicity. The structure $\mathbf{S}_2$ for $\mathbf{S}$ defined in Example 4 is shown in Fig. 3. The nullary and unary predicates are depicted as node labels. The fact that the nullary predicate $N$ holds in $\mathbf{S}_2$ is depicted as a label shared across all the objects. If there are several binary relations connecting two nodes, we list them along a single edge.

Once we have the binary structure $\mathbf{S}_2$, we can build the corresponding GNN input. First, we encode the information about nullary and unary predicates into node feature vectors. This can be done by the multi-hot encoding (Sec. 2), where we treat the nullary predicates as unary ones that either hold for all objects or none. Then, for a graph variant, we encode the information about all the relations connecting two nodes into a respective *multi-hot* edge feature vector or, for a multi-graph variant, we treat each such relation as a separate edge with a corresponding *one-hot* feature vector. Depending on the applied GNN model, the edge feature vectors are then either exploited or ignored.

### Atom Binary Structure

The next natural choice for objects of $\mathbf{S}_2$ is the *atoms* that hold in $\mathbf{S}$. The concept of connecting two atoms is then based on their contained and shared objects. First, we connect two atoms if they share common objects. Second, we also connect two atoms if the symmetric difference of objects occurring in the atoms is related by a relation from $\mathbf{S}$.

Let $\mathcal{L}$ be a language and $m$ the highest arity of a predicate in $\mathcal{L}$. We define a new language $\mathcal{L}_A$ such that each predicate from $\mathcal{L}$ is in $\mathcal{L}_A$ as a unary predicate, and for each pair $\langle i, j \rangle \in [1, m]^2$ there is a binary predicate $E_{i,j}$. Finally, for each predicate $R$ of arity at least 2, there is a binary predicate $\hat{R}$ in $\mathcal{L}_A$.

**Definition 6.** Let $\mathbf{S} = \langle S, \langle R^{\mathbf{S}} \mid R \in \mathcal{L} \rangle \rangle$ be an $\mathcal{L}$-structure. We define its *atom binary structure* as an $\mathcal{L}_A$-structure $\mathbf{S}_2 = \langle A, \langle R^{\mathbf{S}_2} \mid R \in \mathcal{L}_A \rangle \rangle$ where $A$ is the set of all atoms that hold in $\mathbf{S}$, i.e., each atom $\alpha \in A$ is of the form $R(\vec{c})$ for some tuple $\vec{c} \in R^{\mathbf{S}}$.

For a unary predicate $R \in \mathcal{L}_A$ and an atom $\alpha \in A$, we define

$$\alpha \in R^{\mathbf{S}_2} \text{ iff } \alpha = R(\vec{c}).$$

Hence, the unary predicates in $\mathcal{L}_A$ express the predicate of a given atom.

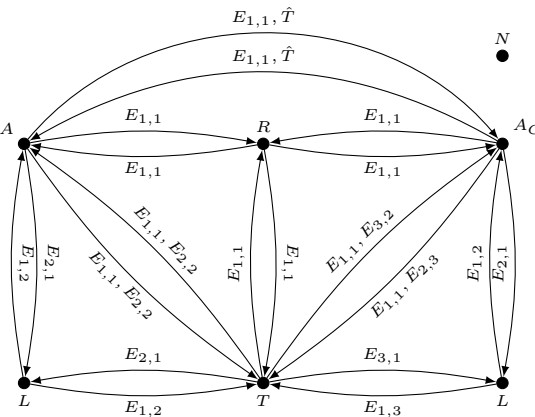

Figure 4: Atom binary structure corresponding to the structure from Example 4.

For a binary predicate $E_{i,j} \in \mathcal{L}_A$ and atoms $\alpha, \beta \in A$, we define

$$\langle \alpha, \beta \rangle \in E_{i,j}^{\mathbf{S}_2} \text{ iff } \alpha = R(\vec{b}),\ \beta = R'(\vec{c}) \text{ and } b_i = c_j.$$

In words, the binary predicates $E_{i,j}$ tell us that the $i$-th object of the atom $\alpha$ is the same as the $j$-th object in the atom $\beta$.

For a binary predicate $\hat{R} \in \mathcal{L}_A$ and atoms $\alpha, \beta \in A$, we define

$$\langle \alpha, \beta \rangle \in \hat{R}^{\mathbf{S}_2} \text{iff } \alpha = R'(\vec{b}),\ \beta = R''(\vec{b}') \text{ and }$$
$$\vec{b} \triangle \vec{b}' \subseteq \vec{c} \text{ for some } \vec{c} \in R^{\mathbf{S}}.$$

The meaning of the predicate $\hat{R}$ is that if we remove all common objects from $\vec{b}$ and $\vec{b}'$, the remaining set of objects appear in an atom of the form $R(\vec{c})$ that holds in $\mathbf{S}$.

The atom binary structure $\mathbf{S}_2$ for the running example is depicted in Fig. 4. Each atom is labeled by its predicate. For example, the atom $T(r, l_1, l_2)$ is labeled by $T$, and $A_G(r, l_2)$ is labeled by $A_G$. Then note the directed edge from $T(r, l_1, l_2)$ to $A_G(r, l_2)$ labeled by $E_{1,1}$ and $E_{3,2}$, as the first object $r$ in $T(r, l_1, l_2)$ is also the first object in $A_G(r, l_2)$. Likewise, the third object $l_2$ in $T(r, l_1, l_2)$ is the second object in $A_G(r, l_2)$. Also note that the atom $A(r, l_1)$ and $A_G(r, l_2)$ are related by $\hat{T}$, since if we remove their shared object $r$, the remaining objects $l_1, l_2$ occur in the atom $T(r, l_1, l_2)$.

To make a GNN input from the atom binary structure, we follow a process analogous to the object binary structure. That is, the unary predicates are encoded into the node features, and the binary relations are encoded into edge features, forming either graph or multi-graph, depending on the choice of multi-hot or one-hot encoding, respectively.

## Object-Atom Binary Structure

The next encoding builds the binary structure $\mathbf{S}_2$ on both objects and atoms. This naturally leads to a bipartite graph as we have two sorts of nodes. The edge relations connect an object with an atom if the object occurs in the atom. Let $\mathcal{L}$ be a language and $m$ the highest arity of a predicate in $\mathcal{L}$. We

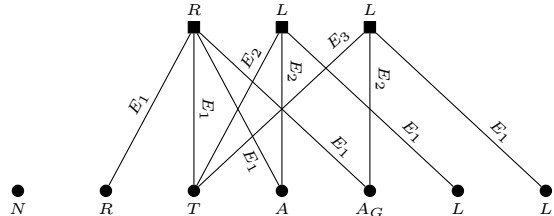

Figure 5: Object-atom binary structure corresponding to the structure from Example 4.

define a new language $\mathcal{L}_{OA}$ such that each predicate from $\mathcal{L}$ is in $\mathcal{L}_{OA}$ as a unary predicate, and for each $i \in [1, m]$ there is a binary predicate $E_i$. Finally, $\mathcal{L}_{OA}$ has two extra unary predicates $O, F$ to distinguish objects from atoms.

**Definition 7.** Let $\mathbf{S} = \langle S, \langle R^{\mathbf{S}} \mid R \in \mathcal{L} \rangle \rangle$ be an $\mathcal{L}$-structure. We define its *object-atom binary structure* as an $\mathcal{L}_{OA}$-structure $\mathbf{S}_2 = \langle S \cup A, \langle R^{\mathbf{S}_2} \mid R \in \mathcal{L}_{OA} \rangle \rangle$ where $A$ is the set of all atoms that hold in $\mathbf{S}$, $O^{\mathbf{S}_2} = S$, and $F^{\mathbf{S}_2} = A$. For a predicate $R \in \mathcal{L}$ and an atom $\alpha \in A$, we define for the unary $R \in \mathcal{L}_{OA}$:

$$\alpha \in R^{\mathbf{S}_2} \text{ iff } \alpha = R(\vec{c}).$$

In other words, each atom knows its predicate. Moreover, if $R \in \mathcal{L}$ is unary and an object $b \in S$, we further define

$$b \in R^{\mathbf{S}_2} \text{ iff } b \in R^{\mathbf{S}}.$$

This means that each object knows its unary properties.

For an object $b \in S$ and an atom $\alpha \in A$, we define

$$\langle b, \alpha \rangle, \langle \alpha, b \rangle \in E_i^{\mathbf{S}_2} \text{ iff } \alpha = R(\vec{c}) \text{ and } b = c_i.$$

In other words, we relate the object $b$ with the atom $\alpha$ via $E_i$ iff $b$ occurs in the $i$-th position within the atom $\alpha$.

The object-atom binary structure for the structure $\mathbf{S}$ from Example 4 is shown in Fig. 5. The atoms (i.e., elements of $A$) are depicted as black dots whereas the objects (i.e., elements of $S$) as black boxes. The atoms are labeled by their predicates and objects by the unary predicates from $\mathcal{L}$ they satisfy. An object is connected to an atom if it appears in that atom. For instance, the object $l_2$ is connected to $T(r, l_1, l_2)$ via $E_3$, to $A_G(r, l_2)$ via $E_2$, and to $L(l_2)$ via $E_1$.

The GNN input is then formed analogously to the previous two cases, forming edge features from the respective binary relations, and two types of feature vectors for the object and atom nodes, respectively, resulting in a bipartite (multi) graph. Thus the information from the feature vectors attached to objects is propagated to the feature vectors of atoms, and vice versa. We note that the multi-graph variant of this bipartite encoding, together with a suitable GNN architecture, roughly corresponds to the model applied in (Ståhlberg, Bonet, and Geffner 2022a,b, 2023).

## Object-Pair Binary Structure

To reach beyond the expressive power of the $C_2$ logic, we also introduce an encoding inspired by the higher-order GNNs, namely 2-GNN (Morris et al. 2019). Given a graph

G, 2-GNN applies the message passing scheme to a graph built on 2-sets of objects. Two 2-sets $\{b_1, b_2\}$ and $\{c_1, c_2\}$ are connected by an edge if $|\{b_1, b_2\} \cap \{c_1, c_2\}| = 1$, i.e., if they share a single object. Morris et al. also consider a "local" variant of the graph where the 2-sets are connected if they share a single object and the objects in their symmetric difference are connected by an edge in **G**.

We model these 2-sets as 2-tuples whose components are sorted by a fixed total order $\leq$ on the set of objects, i.e., $\vec{b} = \langle b_1, b_2 \rangle$ for objects $b_1, b_2$ such that $b_1 \leq b_2$. Using the non-strict $\leq$, unlike Morris et al., we thus also consider the singletons, represented as $\langle b, b \rangle$. Given a finite set $S$, we denote $[S]_2 = \{\vec{b} = \langle b_1, b_2 \rangle \mid b_1, b_2 \in S, b_1 \leq b_2\}$. We refer to the elements of $[S]_2$ as *ordered pairs*.

Let $\mathcal{L}$ be a language. We define a new language $\mathcal{L}_{OP}$. The languages $\mathcal{L}$ and $\mathcal{L}_{OP}$ agree on the nullary predicates. For each unary predicate $P$ in $\mathcal{L}$ we have two copies $P_1, P_2$ in $\mathcal{L}_{OP}$. For the remaining predicates $R \in \mathcal{L}$ of arity at least 2, we have one unary predicate $R$ and one binary $\hat{R}$ in $\mathcal{L}_{OP}$. The language $\mathcal{L}_{OP}$ further has an extra binary predicate $E$.

**Definition 8.** Let $\mathbf{S} = \langle S, \langle R^{\mathbf{S}} \mid R \in \mathcal{L}_{OP} \rangle \rangle$ be an $\mathcal{L}$-structure. We define its *object-pair binary structure* as an $\mathcal{L}_{OP}$-structure $\mathbf{S}_2 = \langle [S]_2, \langle R^{\mathbf{S}_2} \mid R \in \mathcal{L}_{OP} \rangle \rangle$ where $S_2$ is the set of all ordered pairs of objects in $S$. The interpretation of nullary predicates in $\mathbf{S}_2$ is the same as in $\mathbf{S}$. For a unary predicate $P \in \mathcal{L}$ and an ordered pair $\vec{b} = \langle b_1, b_2 \rangle \in [S]_2$, we define

$$\vec{b} \in P_1^{\mathbf{S}_2} \text{ iff } b_1 \in P^{\mathbf{S}}.$$

and similarly,

$$\vec{b} \in P_2^{\mathbf{S}_2} \text{ iff } b_2 \in P^{\mathbf{S}}.$$

For a unary predicate $R \in \mathcal{L}_{OP}$ corresponding to a predicate $R \in \mathcal{L}$ of arity at least 2 and an ordered pair $\vec{b} = \langle b_1, b_2 \rangle \in [S]_2$, we define

$$\vec{b} \in R^{\mathbf{S}_2} \text{ iff } \{b_1, b_2\} \subseteq \vec{c} \text{ for some } \vec{c} \in R^{\mathbf{S}}.$$

In other words, the ordered pairs know if their inner objects are related by the predicate $R$. For two ordered pairs $\vec{b}, \vec{c} \in [S]_2$, we define

$$\langle \vec{b}, \vec{c} \rangle \in E^{\mathbf{S}_2} \text{ iff } \vec{b} \neq \vec{c} \text{ and } |\vec{b} \cap \vec{c}| = 1.$$

Lastly, for a binary predicate $\hat{R} \in \mathcal{L}_{OP}$ and two ordered pairs $\vec{b}, \vec{c} \in [S]_2$, we define

$$\langle \vec{b}, \vec{c} \rangle \in \hat{R}^{\mathbf{S}_2} \text{ iff } \langle \vec{b}, \vec{c} \rangle \in E^{\mathbf{S}_2} \text{ and } \vec{b} \triangle \vec{c} \subseteq \vec{d} \text{ for some } \vec{d} \in R^{\mathbf{S}}.$$

The object-pair binary structure for the structure $\mathbf{S}$ from Example 4 is shown in Fig. 6. Note that the validity of the nullary atom $N$ is depicted as a label $N$ attached to all ordered pairs. The predicates labeling the ordered pairs with the subscript 1 (resp. 2) refer to properties of the first (resp. second) object in the pair. Ordered pairs are also labeled by the predicates relating their components. For instance, the ordered pair $\langle r, l_2 \rangle$ is labeled by $A_G$ and $T$ because $r, l_2$ occur in $A_G(r, l_2)$ and $T(r, l_1, l_2)$. The edges labeled by $E$ connect different ordered pairs sharing a common object, e.g., $\langle l_1, l_1 \rangle$ is related to $\langle l_1, l_2 \rangle$ by $E$. Further edges connect

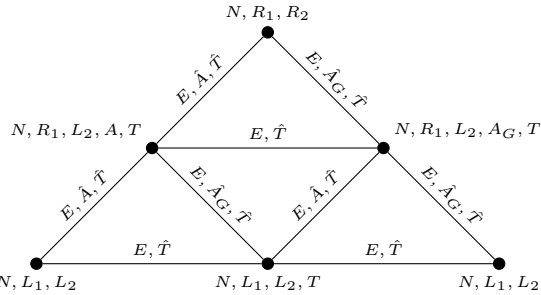

Figure 6: Object-pair binary structure corresponding to the structure from Example 4.

those ordered pairs, already connected by $E$, whose symmetric difference-objects are related by a predicate. For instance, there is an edge $\hat{T}$ between $\langle r, l_1 \rangle$ and $\langle r, l_2 \rangle$ because they are related by $E$, $\langle r, l_1 \rangle \triangle \langle r, l_2 \rangle = \{l_1, l_2\}$ and both $l_1, l_2$ occur in $T(r, l_1, l_2)$.

The GNN input is created from the $\mathcal{L}_{OP}$-structure $\mathbf{S}_2$ in the same way as for the object binary structure.

## 4 Expressiveness

The previous section detailed several encodings transforming an enriched state to a data structure suitable for existing GNN models. In this section, we take a theoretical point of view on their expressive power. Recall that we are interested in whether a PDDL domain $\mathcal{D}$ contains bad indistinguishable pairs of enriched states (Sec. 1). Since the policy algorithm is supposed to solve instances within a given $\mathcal{D}$, we only need to distinguish pairs of states from a single instance, which are always defined on a fixed set of objects and enriched by the same goal.

Firstly, the expressive power of the algorithm might be limited by the encoding. In practice (Sec. 5), this is often due to the conversion of the arbitrary input structure to the binary one. For instance, consider a language $\mathcal{L}$ with a single ternary predicate $T$ and an $\mathcal{L}$-structure $\mathbf{S} = \langle \{0, 1, 2\}, T^{\mathbf{S}} \rangle$ where $T^{\mathbf{S}} = \{\langle 0, 1, 1 \rangle, \langle 1, 2, 2 \rangle, \langle 2, 0, 0 \rangle\}$. The object binary structure for $\mathbf{S}$ (Definition 5) is then just a triangle with self-loops on the set $\{0, 1, 2\}$. Now consider a similar $\mathcal{L}$-structure $\mathbf{S}'$ whose interpretation of $T$ is $T^{\mathbf{S}'} = T^{\mathbf{S}} \cup \{\langle 0, 1, 2 \rangle\}$. Apparently, if we create the object binary structure for $\mathbf{S}'$, we end up with the same triangle as before. Thus $\mathbf{S}$ and $\mathbf{S}'$ are indistinguishable if we apply the encoding from Definition 5. On the other hand, the encodings given by the atom binary (Definition 6) and object-atom binary (Definition 7) structures can distinguish $\mathbf{S}$ and $\mathbf{S}'$, as these structures differ in the number of valid ground atoms, corresponding to nodes.

Secondly, the expressiveness of a GNN is primarily limited by the $C_2$-equivalence (Theorem 2) w.r.t. its input graphs. Hence, if the graphs produced by the encoding for a pair of enriched states $\mathbf{S}$ and $\mathbf{S}'$ are $C_2$-equivalent, the GNN will predict the same output for both. For instance, consider the well-known "blocksworld" PDDL domain, which contains at most binary predicates, ensuring that none of

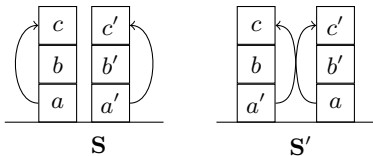

Figure 7: Two $C_2$-equivalent states in the blocksworld

the introduced encodings loses information. Nevertheless, one can construct $C_2$-equivalent blocksworld states $\mathbf{S}, \mathbf{S}'$ such that $V^*(\mathbf{S}) \neq V^*(\mathbf{S}')$, an example if which is shown in Fig. 7. The goal for both states is defined by $\psi_G = \{On_G(c,a), On_G(c',a')\}$, as depicted by arrows. The structure $\mathbf{S}$ satisfies the following $C_3$-sentence, while $\mathbf{S}'$ does not:

$$\exists x, y, z(On(y,x) \wedge On(z,y) \wedge On_G(z,x))$$

This sentence expresses that there is a "cycle" of length 3 that cannot be equivalently expressed in $C_2$. If we encode $\mathbf{S}, \mathbf{S}'$ by the object binary structure (Definition 5), we obtain just their symmetric versions that are $C_2$-equivalent as well. Similarly, the encoding based on the object-atom binary structure (Definition 7) does not resolve the issue, as it also produces $C_2$-equivalent graphs.

Nevertheless, we can overcome the limits of $C_2$-equivalence for the above-mentioned blocksworld states using the object-pair binary structure encoding (Definition 8). Recall that this encoding forms ordered pairs of objects that are connected if they share a common object. This shared object can then be exploited as a "memory" for detecting the cycles. More precisely, the object-pair binary structure for $\mathbf{S}$ satisfies the following $C_2$-sentence:

$$\exists x, y(On(x) \wedge On_G(y) \wedge \widehat{On}(x,y))$$

Note that the variables $x, y$ are now interpreted as ordered pairs. The witnesses showing that the above sentence holds are the pairs $\langle a, b \rangle$ and $\langle a, c \rangle$. This yields $On(\langle a, b \rangle)$, $On_G(\langle a, c \rangle)$, and $\widehat{On}(\langle a, b \rangle, \langle a, c \rangle)$ as these pairs share the object $a$, and the remaining objects $b, c$ are related by $On$ in $\mathbf{S}$. On the other hand, the object-pair encoding of $\mathbf{S}'$ does not satisfy the above sentence since none of the ordered pairs sharing a single object is related by $\widehat{On}$.

Note that similar reasoning can also be applied to the atom binary structure (Definition 6) since each atom, in fact, represents a tuple of objects. Therefore it effectively distinguishes the $C_2$-equivalent blocksworld states as well. However, the collection of atoms need not represent all $k$-tuples of objects (e.g., all the ordered pairs). Consequently, the atom binary structure and object-pair binary structure encodings are not directly comparable in their expressive power.

## 5 Experimental Study

The theoretical investigation of expressiveness for all the particular encodings and GNNs is considerably challenging, given the plethora of existing models and the limited applicability of Theorem 2 and Theorem 3 to only a selected subset (Xu et al. 2019). To efficiently address this challenge, we propose an alternative, experimental protocol based on testing whether a given encoding-GNN integration correctly recognizes bad indistinguishable pairs in a set of generated planning states. To conduct this evaluation, we consider IPC domains. For each domain and each of its instances, we then generate random states and label them with the value of $V^*$, computed by a planner (if a plan is found).

Recall that each combination of an encoding and a GNN model yields a function $f$, assigning a prediction $f(\mathbf{S}, \theta)$ to each state structure $\mathbf{S}$ given some GNN parameters $\theta$, and that for any indistinguishable pair of states $\mathbf{S}, \mathbf{S}'$ it must hold $f(\mathbf{S}, \theta) = f(\mathbf{S}', \theta)$ for any $\theta$ (Sec. 1). Thus, if $f(\mathbf{S}, \theta) \neq f(\mathbf{S}', \theta)$ for some $\theta$, we may instantly refute the pair from being bad. On the other hand, if the states yield an equal value, it may still be due to chance rather than being indistinguishable. Note, however, that $f$ is continuous in $\theta$, as we commonly use injective activation functions, such as $\tanh$,[2] and most of the sampled parameters in $\theta$ are non-zero. Consequently, the probability of this happening is rather small, which we further reduce by the "amplification trick" used in design of randomized algorithms (Sourek, Zelezny, and Kuzelka 2020). Specifically, we repeatedly sample $\theta$ and evaluate all the states in a given set to yield a list of predictions $(f(\mathbf{S}_i, \theta_1), \ldots, f(\mathbf{S}_i, \theta_{rep}))$ for each $\mathbf{S}_i$. Using a reasonable combination of the prediction value numeric precision ($digits = 6$) and number of repeated $\theta$-evaluations ($rep = 3$), we effectively detect all the indistinguishable subsets of states. Note that this can be done efficiently in linear amortized time using a hash table. The resulting number of "bad" state pairs then comes from combining the indistinguishable subsets with the $V^*$-equivalent subsets, and reporting the sum of their $C(n,2)$ combinations.

Tab. 1 contains the results of our analysis. The considered encodings are **O**, **A**, **O-A**, and **O-P**, corresponding to the object, atom, object-atom, and object-pair binary structures, respectively. The superscripts **G**, **MG**, **BG**, and **BMG** then denote whether the resulting GNN encoding is a graph, multigraph, bipartite graph, or bipartite multigraph, respectively. None of the **A** encodings use the $\hat{R}$ predicates (see Definition 6) except for $\hat{\mathbf{A}}^{\text{MG}}$.

The selection of the presented GNN models for the experiments was based on their practical prevalence and compatibility with the input structures, ensuring comprehensive support for the respective edge features, multi-graphs, and bipartite graphs. All their implementations come from the standardized PyTorch Geometric library (Fey and Lenssen 2019), where they are available under the reported names. We performed ablation w.r.t. their parameter dimensionalities, number of layers, and aggregation operators. We found the dimensionalities to be of negligible statistical importance beyond a reasonably low value ($dim = 16$), and similarly for the number of layers ($layers = 8$), corresponding to the iterations of the CR (Sec. 2). The choice of the aggregation has a significant impact though, with $aggr = sum$ being clearly superior to $mean$ and to $max$, respectively, in accordance with the findings in (Xu et al. 2019).

All the selected PDDL domains come from IPC, with the

---

[2]This is slightly complicated by the fixed ReLU functions used in some of the GNNs, but is of negligible practical impact.

| Model / Encoding | $O^G$ | $O^{MG}$ | $O\text{-}A^{BG}$ | $O\text{-}A^{BMG}$ | $A^G$ | $A^{MG}$ | $\hat{A}^{MG}$ | $O\text{-}P^G$ | $O\text{-}P^{MG}$ |
|---|---|---|---|---|---|---|---|---|---|
| GCN (Kipf and Welling 2016) | 222,04 | 30,09 | | | | | | | |
| SAGE (Hamilton, Ying, and Leskovec 2017) | 249,78 | 32,66 | 54,80 | 61,94 | 71,72 | 4,54 | 21,80 | 340,03 | 31,48 |
| GIN (Xu et al. 2019) | 227,46 | 24,55 | | | 142,20 | 35,05 | 42,41 | 18,49 | 17,73 |
| GINE (Hu et al. 2020) | 12,22 | 61,33 | | | 40,19 | 4,55 | **1,67** | 26,46 | 23,31 |
| GATv2 (Brody, Alon, and Yahav 2021) | 32,56 | 22,03 | 99,19 | 78,21 | 50,85 | 6,38 | 4,18 | 7,64 | 6,27 |
| NN (Gilmer et al. 2017) | 29,40 | 38,16 | | | 44,26 | 3,72 | **1,03** | 5,34 | **1,73** |
| Transformer (Shi et al. 2021) | 34,55 | 30,44 | 102,30 | 66,82 | 49,12 | 3,64 | **1,65** | 12,66 | 9,89 |
| PDN (Rozemberczki et al. 2021) | 23,04 | 19,47 | | | 45,38 | 6,56 | 2,48 | 43,08 | 17,95 |
| GEN (Li et al. 2020) | 29,46 | 32,06 | 54,83 | 65,96 | 53,74 | **1,75** | **2,05** | **1,30** | **1,00** |
| General (You, Ying, and Leskovec 2020) | 31,79 | 39,59 | 52,25 | 49,12 | 46,39 | 3,57 | 3,21 | 337,92 | 32,92 |
| RGCN (Schlichtkrull et al. 2018) | 212,42 | 44,64 | 62,70 | 51,92 | | | | | |
| FiLM (Brockschmidt 2020) | 233,12 | 27,91 | 59,93 | 48,43 | | | | | |
| **Domain / Encoding** | | | | | | | | | |
| barman | **0,00** | **0,00** | **0,00** | **0,00** | **0,00** | **0,00** | **0,00** | **0,00** | **0,00** |
| blocks | 0,31 | 0,31 | **0,00** | **0,00** | **0,00** | **0,00** | **0,00** | 0,04 | 0,04 |
| depot | **0,00** | **0,00** | 0,13 | 0,13 | **0,00** | **0,00** | **0,00** | 0,13 | 0,13 |
| elevators | **0,00** | **0,00** | **0,00** | **0,00** | 0,06 | **0,00** | **0,00** | **0,00** | **0,00** |
| floortile | **0,00** | **0,00** | **0,00** | **0,00** | **0,00** | **0,00** | **0,00** | **0,00** | **0,00** |
| freecell | **0,00** | **0,00** | **0,00** | **0,00** | **0,00** | **0,00** | **0,00** | **0,00** | **0,00** |
| gripper | **0,00** | **0,00** | **0,00** | **0,00** | **0,00** | **0,00** | **0,00** | **0,00** | **0,00** |
| logistics | **0,00** | **0,00** | 0,15 | 0,25 | **0,00** | **0,00** | **0,00** | 0,05 | 0,30 |
| parcprinter | 59,83 | 60,17 | 57,83 | 57,83 | 61,83 | 84,50 | 77,00 | 7,50 | 7,33 |
| pegsol | **0,00** | **0,00** | **0,00** | **0,00** | **0,00** | **0,00** | **0,00** | **0,00** | **0,00** |
| rovers | **0,00** | **0,00** | 0,04 | 0,04 | 0,17 | 0,55 | 0,55 | **0,00** | **0,00** |
| satellite | **0,00** | **0,00** | **0,00** | **0,00** | **0,00** | **0,00** | **0,00** | 0,21 | 0,79 |
| scanalyzer | 87,00 | 153,50 | 6,75 | 8,00 | 27,75 | 1,25 | 27,75 | 6,50 | 6,00 |
| sokoban | 644,55 | 695,85 | 1243,15 | 1508,00 | 1209,35 | 8,95 | 19,26 | 1,00 | 0,67 |
| tidybot | 0,05 | 0,05 | 0,05 | 0,05 | **0,00** | **0,00** | **0,00** | 0,10 | **0,00** |
| tpp | **0,00** | 0,18 | **0,00** | **0,00** | **0,00** | **0,00** | **0,00** | **0,00** | **0,00** |
| transport | **0,00** | **0,00** | **0,00** | 0,08 | **0,00** | **0,00** | **0,00** | **0,00** | 0,17 |
| visitall | 0,07 | **0,00** | 0,20 | 0,07 | **0,00** | **0,00** | **0,00** | **0,00** | **0,00** |
| woodworking | **0,00** | **0,00** | **0,00** | **0,00** | **0,00** | **0,00** | **0,00** | **0,00** | **0,00** |
| pipesworld | **0,00** | **0,00** | **0,00** | **0,00** | **0,00** | **0,00** | **0,00** | **0,00** | **0,00** |
| vacuum-sep | 1,50 | 1,10 | 4,95 | 4,10 | 4,95 | 1,35 | **0,00** | 5,20 | 3,55 |
| vacuum-sh | 3,75 | 0,40 | 7,40 | 4,65 | 10,60 | 2,65 | 0,20 | 13,80 | 10,10 |

Table 1: Average number of bad indistinguishable state pairs (Sec. 1) across the introduced encodings (Sec. 3) and common GNN models (upper part); and across common planning domains (lower part) using a fixed GNN model (GEN by Li et al.). The bold numbers in the upper part denote the best-performing 10% of configurations.

exceptions of "vacuum-sep" and "vacuum-sh" introduced in (Ståhlberg, Bonet, and Geffner 2022a). These domains model multiple robots cleaning a single spot. The two variants differ in the maps defining the robots' allowed moves. In the first variant, each robot has its own map, unlike in the second variant, where the map is shared across the robots.

The upper part of Tab. 1 presents the average numbers of bad state pairs for a particular encoding and GNN model across all PDDL domains. The absence of certain combinations is due to the incompatibility of the GNN model with the respective encoding. Notably, the combinations based on the most expressive encodings of the atom and object-pair binary structures (Sec. 3) tend to perform the best overall. The lower part of the table displays the performance of the encodings for the domains using a fixed GNN model (GEN by Li et al.). As can be seen, for each domain there exists an encoding that can almost perfectly distinguish the states, with exceptions for "parcprinter", "scanalyzer", "sokoban", and both the "vacuum" domains. Interestingly, the "sokoban" domain appears particularly challenging across a wide range of settings, but can be effectively addressed with the **A, O-P** encodings. We note that

a common trait of the difficult domains is the use of predicates of arity greater than 2. Conversely, the "pegsol" domain employs a ternary predicate and still exhibits no bad pairs. However, the ternary predicate in "pegsol" is static (unaffected by actions), and the states are modeled by unary predicates, making the domain likely more manageable.

## 6 Conclusions

We systematically defined possible encodings of planning states into data structures compatible with common GNNs. Each such proposed encoding, in conjunction with several compatible GNN models, was assessed for the presence of undesirably indistinguishable state pairs. Our findings highlight the superior expressiveness of the encodings based on atoms and object-pairs, relative to the, more common, encodings based on object-atom and object binary structures. Nevertheless, their superiority is accompanied by larger GNN inputs and runtimes. Most of the empirical limitations of GNN expressiveness in planning domains then seem to stem from the presence of predicates with arity exceeding 2, a conclusion that aligns intuitively with the binary nature of the message-passing scheme underlying the GNNs.

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
