# OpenReview forum: "Expressiveness of Graph Neural Networks in Planning Domains"
_icaps-conference.org/ICAPS/2024/Conference — ICAPS 2024_

### Official Review · Reviewer_yULi · 2024-01-16

**Significance And Importance:** 3
**Soundness:** 3
**Novelty:** 3
**Clarity:** 4
**Overall Evaluation:** 2
**Confidence:** 3

**Weaknesses:**

1: Minor weaknesses that are easily fixable.

**Contributions Of The Paper:**

The authors study the problem of learning a value function for classical deterministic planning. They are interested in developing and studying graphical encodings of planning states for GNN models of the optimal value function in particular. Functions represented as GNNs have a familiar Achilles' heel. For planning specifically we have that there are pairs of states whose optimal values are distinct, but which cannot be distinguished by GNNs. The authors note that this issue can be mitigated by judicious choices regarding: how states are represented graphically and the GNN model details and parameters. Generally GNNs can never be sufficiently expressive to distinguish all pairs of states that should be distinguished as having distinct values. Such models are nonetheless powerful/useful, and pushing the envelope on how far these can be stretched is of immense interest. The submitted work is thoughtful, detailed, and includes a systematic empirical study of the distinguishability problem in the context of optimal value functions over a wide range of interesting encodings and models. This is a new study about a contemporary method, shedding new light on GNNs specifically in regards to how they might be used in an optimal planning setting.

**Ethical Considerations:**

(1) Not Applicable: The paper does not have any ethical considerations to address

**Nomination For Best Paper:**

No

**Questions For Authors:**

Why are you confident your experimental datasets comprise a fair and representitive set of states wrt the opportunity for badness in each domain?

**Reproducibility:**

3: Authors describe the implementation and domains in sufficient detail.

**Strengths Of The Paper:**

Beyond the empirical work a substantial contribution here is the description of 4 graphical encodings of states. There is a great minimalistic running problem example also used to showcase different aspects of the described encodings.  Definitions 5 and 6 are presented, it seems, as relatively straightforward baseline encodings for the sake of contrast with Definitions 7 and 8. The "object-atom binary structure" (Definition 7) encoding explicitly exposes information to the GNN about objects' in ground predicates, and  I understand is intended as a representation of existing encodings from the very recent literature. The "object-pair binary structure" state encoding I understand is a new contribution, intended to stretch GNNs beyond Def. 7. It seems to be clearly motivated by the empirical results as interesting for mitigating "bad" indistinguishability in practice. Along with definitions of new encodings, an analytical case study using a blocks world instance (see Section 4) is described to motivate and characterise aspects and limitations of some provided encodings. For example, we are shown how Def. 8 can mitigate indistinguishability experienced by some other encodings.

A substantial contribution of the paper is the empirical evaluation over: (i) a large range of GNN models, (ii) all encodings described and developed in the paper along with some variants (wrt type of graph used to represent states) of those, and (iii) an appropriately comprehensive set of PDDL planning benchmarks (I noted including some with exploitable symmetry). The authors are able to clearly showcase encodings and models that exhibit minimal "bad" indistinguishability in a range of domains, and additionally break this down for the reader in terms of specific domains. The contrast between encodings and models that work (wrt distinguishing states with distinct reward values) against those that do not is quite high.

**Weaknesses Of The Paper:**

The lack of detail explicitly given in setting up and describing the data presented about the experimental work is the biggest weakness. Questions I had regarding this that I did not see answered were:

 - What are the state(/problem) sizes (#states) and numbers of objects?

 - Line 590, authors mention they generate "random states". How are these generated? Are they states reachable from the initial state of a benchmark task? Or are they truly random, and potentially 'invalid' in the sense they would never be encountered when solving a planning task? Does any care need to be taken to ensure these states are somehow representative of a relevant and sufficiently sized set wrt badness hunting/measuring? What about the distributions of values of the random states encountered in different domains? Do these all have the same shapes?

Note: The above two points should be explained more in the paper, but given things are assumed to be somehow fair across all encoding-model measurements, I do not consider this a huge weakness.

 - Reported is the average number of bad pairs. This admits comparisons as per the paper, because we suppose all encoding-models are evaluated here over the same set. So things are acceptable. But the weakness is that it's challenging to interpret the provided average, and get a sense of whether GNNs are likely or unlikely to be 'bad' in the sense developed here. I do not have a sense of the opportunity for badness or how that scales in different domains.

 - Actual function approximation performance at value function modelling and computation time are not considered.

 - On Line 613 the authors chose Rep=3. It is not obvious why they are confident that this number of repetitions is sufficient.

 - To reproduce exactly this experimental data may be quite difficult (inc. details related to the specifics of orderings for Def 8.), and it would be helpful if the software that is specific to planning, datasets of state-values (or for generating the same as used here), and these experiments were available to the community. Overall however, it seems the core distinguishing features of the state representations for the different encodings are described in sufficient detail. So, we should expect to reproduce the main experimental conclusions.

More detail about the different graph types would be welcome/helpful, although the lack of space in this format is noted as a constraining on this point.

On Line 100 -  I think finiteness conditions be mentioned in this paragraph.

grammar: "The counting quantifiers are definable in FOL, but they are not if we restrict the number of variables."

---

> ### Author Rebuttal · Authors · 2024-01-26
>
> We thank the reviewer for valuable comments and remarks.
>
> Q1) We originally considered generating only reachable states from the initial states occurring in the problem files by random walk like Stahlberg et al. (2022a). However, that would be biased as we want to test the GNN against the whole domain, i.e., we can look for a plan from any reasonable initial state, not only those appearing in the IPC datasets. Moreover, the random walk does not ensure uniform distribution.
>
> On the other hand, generating a random state in the PDDL representation might easily produce a nonsensical state, as suggested by the reviewer, like a truck being at two locations simultaneously. Hence, we decided to convert each planning task into SAS+. This eliminates many of the nonsensical states. Some random states in SAS+ might also be nonsensical, like in Blocks World. However, as we consider only states having a plan, these are mostly eliminated. Thus, any considered state could likely be encountered when solving a domain task with a regular planner, rendering the resulting sets representative of the standard setting, with no special measures taken w.r.t. the “badness hunting”.
>
> More precisely, the datasets were generated as follows. For each domain, we considered all the available problem files. Each of them was converted into SAS+. For the SAS+ representation, we uniformly generated a random state and tried to find a plan by the Fast Downward planner with a timeout limit. The state was discarded if no plan was found; otherwise, all the states along the plan were added to the domain dataset. This process was repeated until more than 100 states were generated or a global timeout was reached. On average, we had 70 states per problem file. We will update Table 1 with each domain's average number of states and objects per instance and their average V* values.
>
> The code and the datasets will be made available on Github upon deanonymization.
>
> The choice of rep=3 (in conjunction with digits=6) was made with an extra (computationally demanding) test of pair-wise similarity between all the states’ outputs, ensuring that no non-distinguishable state pairs pass the routine as separate while keeping the distinguishable ones apart. However, the chance of states being incorrectly identified as indistinguishable in the continuous space of the GNN functions is extremely small even with rep=1.
>
> Please see the answer for LRn2 regarding learning V* and sD8V regarding the runtimes, respectively.

---

### Official Review · Reviewer_sD8V · 2024-01-22

**Significance And Importance:** 2
**Soundness:** 3
**Novelty:** 3
**Clarity:** 3
**Overall Evaluation:** 2
**Confidence:** 4

**Weaknesses:**

2: No major or minor weaknesses.

**Contributions Of The Paper:**

The paper presents an evaluation of different encodings and the expressive power of SOTA GNNs using such encodings. The overall expressive power of a GNN is determined by (a) the encoding, and (b) the expressive power of the GNN which is known to be C_2. If the inference procedure of the GNN yields the same output for two different states then they are said to be indistinguishable. This is a problem in planning since often times we encounter states where we would like two states to be distinguishable.

The authors then propose several different input encodings that can be used to enrich grounded states and these enriched states are suitable as inputs to GNNs. To this effect, the authors propose Object Binary, Atom Binary, Object-Atom Binary and Object-Pair Binary Structure. These encodings are primarily evaluated w.r.t. whether the enriched states can represent concepts that grounded states using GNNs cannot due to their limited expressivitiy to C_2 logic.

The authors conduct an extensive empirical evaluation with several SOTA GNN implementations of different architectures with the encodings on benchmark IPC domains and contrast the performance of the encodings w.r.t. the ability to minimize the # of bad indistinguishable state pairs.

**Ethical Considerations:**

(5) Excellent: The paper comprehensively addresses all of the applicable ethical considerations

**Nomination For Best Paper:**

No

**Questions For Authors:**

1. Could you provide a brief overview of the drawbacks of the Object-Pair Binary Structure w.r.t scalability as the # of objects increase?

Post-rebuttal
===========
Thank you for answering my questions. Very interesting work and will help additional research in generalization.

**Reproducibility:**

3: Authors describe the implementation and domains in sufficient detail.

**Strengths Of The Paper:**

1. The paper is written clearly and the ideas are well-presented.
2. The encodings proposed are interested and will help foster additional research in the area.
3. The empirical evaluation is quite convincing

**Weaknesses Of The Paper:**

1. A contrast with runtimes and added complexity would help provide a better picture of the trade-offs in the paper. The paper makes note of this but does not provide any such metrics.

---

> ### Author Rebuttal · Authors · 2024-01-26
>
> We thank the reviewer for valuable comments and remarks.
>
> Naturally, the runtimes generally reflect the complexity of the resulting encoding graphs, with the object binary structure being generally the fastest, and the object-pair encoding being the slowest. Likewise, there are also (significant) differences in runtimes of the GNN models themselves, with GCN being the fastest and NN(Conv) and RGCN being among the slowest. We will further update Table 1 with aggregated runtime information.
>
> Q1) Naturally, the size of the object-pair representation grows combinatorially (quadratically for nodes and quartically for edges) with the number of domain objects, rendering it unsuitable for large domains (app. > 40 objects), while possibly still not being expressive enough. This is where the (planning-specific) atom-based representation might be more appropriate as the existing atoms, essentially also forming combinations (pairs and beyond) of objects, represent only a particular subset of the combination space, presumably more relevant for the task (than the non-existent atoms/object-combinations).

---

### Official Review · Reviewer_LRn2 · 2024-01-23

**Significance And Importance:** 2
**Soundness:** 3
**Novelty:** 3
**Clarity:** 4
**Overall Evaluation:** 2
**Confidence:** 4

**Weaknesses:**

1: Minor weaknesses that are easily fixable.

**Contributions Of The Paper:**

This paper considers the use of graph neural networks (GNNs) for solving planning problems. In this setting, a GNN maps the encoding of a state and produces an output corresponding to the predicted length of an optimal plan.

The work presents two main contributions: the first one is an extensive set of possible state encodings, which are then transformed into a network amenable as a GNN input. The hope is that the encoding overcomes the limitations in expressive power of GNNs and provides "more informative" features.

The second contribution is an analysis of the expressive power of such encodings, based on the expressive power of the resulting GNNs in finding pairs of states which are "bad indistinguishable" (their GNN output is the same when it should not). The analysis is done theoretically and experimentally.

------------
POST-REBUTTAL: I am largely satisfied with the responses and would like to congratulate the authors for their work.

**Ethical Considerations:**

(1) Not Applicable: The paper does not have any ethical considerations to address

**Nomination For Best Paper:**

No

**Questions For Authors:**

(1) The authors focus on detecting bad indistinguishable pairs. Can they elaborate/motivate why and to what extent is this task more appropriate than focusing on learning the "correct" V*, which is not directly addressed in the paper? Related to this, why not analyzing as well the problem of producing distinct outputs for state pairs which should produce the same output?

(2) How are exactly the GNN parameters chosen in the experimental study? I am assuming that the GNNs are not optimized (trained). Wouldn't it make more sense to compare the outputs of the trained GNNs instead of using sample parameters?

(3) The most recent GNN variants (e.g. subgraph GNNs) overcome "efficiently" the theoretical limits in expressive power derived for "vanilla" GNNs. Is that the case for any of the methods compared in Table 1? Can the authors elaborate a bit more on the implications of losing the theoretical connection with the WL test, and the relevance/significance of their results?

(4) Choosing the identity function as 'agg' in Eq (1) is not permutation invariant. Isn't this a problem to compute a node-coloring as in the CR algorithm?

**Reproducibility:**

2: Some details are missing, but the paper still appears to be replicable with some effort.

**Strengths Of The Paper:**

The paper considers a very relevant line of work. Contrary to many deep learning models, GNNs are well understood theoretically, which is crucial to understand how can they be useful in planning.

The paper is clearly written and the set of proposed encodings are sensible and principled.

The paper draws interesting conclusions favoring encodings based on atoms and object-pairs instead of the more frequently used en
codings based on object-atom and object binary structures.

**Weaknesses Of The Paper:**

The significance of the paper may be not high, given the nature and the scope of the results.
The theoretical analysis of expressiveness (section 4) is rather limited.

minor:
- footnote 1: "\calL do not" -> "\calL does not"
- "an example if which" -> "an example of which"
- there is only one example, should be labelled Example 1, not 4.
- the proofs of the Theorems are missing.

---

> ### Author Rebuttal · Authors · 2024-01-26
>
> We thank the reviewer for valuable comments and remarks.
> Q1)
> a) The problem of detecting the indistinguishability is not "more appropriate" than the overarching task of learning V*, but forms a necessary condition for it.
> We believe that studying this subproblem in isolation, as motivated at the beginning of the paper, then provides a more substantiated insight, from the expressiveness point of view, than simply directly measuring the learning performance of V*, in which a number of other (noisy) factors influence the results.
>
> b) The complementary problem of producing unnecessarily distinct outputs is also very relevant. Nevertheless, such situations will generally be domain-specific, as the isomorphisms form the most general class of symmetries in planning. For two non-isomorphic states, one can always define a planning domain where these states have different V* values.
>
>
> Q2) The GNN parameters are randomly initialized w.r.t. their default distributions (kaiming-uniform, mostly), as they normally would be prior to training (in PyTorch). The alternative of comparing the outputs after training relates directly to Q1.a) of why we do not evaluate the learning of V* instead. Any two states that are indistinguishable prior to training will remain indistinguishable after training. Likewise for distinguishable states, as long as the parameterization is not "unfortunate" (for which we use the repeated sampling). During training, the GNN parameters might produce additional (non-essential) approximate symmetries, mapping the distinguishable states to the same outputs (relating to Q1.b), depending on various factors.
>
> Q3) None of the ("vanilla") GNN models tested in Table 1 exploits subgraphs or higher-order structures on their own. Nevertheless, some of the introduced encodings, particularly those based on atoms and object-pairs, may be seen as extending beyond the C2 expressiveness (over the base object representation). Consequently, the respective GNN-encoding combinations may be seen as such (planning-specific) extensions, reaching beyond the fundamental expressiveness of the 1-WL test, in a fashion similar to that of higher-order (object-pairs) or subgraph (object-atom and atom-atom) GNNs.
>
> Q4) The “identity” function in the description of the CR algorithm acts over multisets. Thus, it is automatically permutation invariant as permuting the elements does not change the resulting multiset.

---

### Meta-Review · Area_Chair_qHzN · 2024-02-03

**Recommendation:** Accept (Oral)
**Confidence:** 4

**Metareview:**

There is a clear consensus to accept this paper. Congratulations. Please take into account the comments on the reviews when preparing the final version.

**Ethical Considerations:**

(1) Not Applicable: The paper does not have any ethical considerations to address